# Development of monoclonal antibodies against E1 protein of Chikungunya virus

**Kun Wen**[1,2☯], **Zhihong Zhou**[1,2☯], **Peipei Xu**[1,2☯], **Biao Di**[3☯], **Jialing Song**[1,2], **Cuilian Yang**[4], **Lili Zhan**[1,2], **Chongquan Zhao**[1,2], **Mei Xu**[1,2], **Jianpiao Cai**[5], **Shuofeng Yuan**[1,2,5,6], **Kelvin Kai-Wang To**[5,6,7], **Wenxi Feng**[8*], **Jasper Fuk-Woo Chan**[5,6,7*], **Hongwei Zhou**[1,2*], **Yue Chen**[9*], **Xixia Ding**[1,2*]

1 GuangDong Engineering Technology Research Center of Rapid Detection for Pathogenic Microorganisms, Zhujiang Hospital, Southern Medical University, Guangzhou, People's Republic of China, 2 Microbiome Medicine Center, Department of Laboratory Medicine, Zhujiang Hospital, Southern Medical University, Guangzhou, People's Republic of China, 3 Center for Disease Control and Prevention of Guangzhou, Guangdong, People's Republic of China, 4 Department of Blood Transfusion, Zhujiang Hospital, Southern Medical University, Guangzhou, People's Republic of China, 5 State Key Laboratory of Emerging Infectious Diseases, Carol Yu Centre for Infection, Department of Microbiology, School of Clinical Medicine, Li Ka Shing Faculty of Medicine, The University of Hong Kong, Pokfulam, Hong Kong, People's Republic of China, 6 Department of Infectious Diseases and Microbiology, The University of Hong Kong-Shenzhen Hospital, Shenzhen, People's Republic of China, 7 Department of Microbiology, Queen Mary Hospital, Pokfulam, Hong Kong Special Administrative Region, People's Republic of China, 8 Shunde District Center for Disease Control and Prevention, Foshan city, People's Republic of China, 9 Center for Disease Prevention and Control of Southern Theatre Command of PLA, Guangdong Arbovirus Diseases Emergency Technology Research Center, Guangzhou, People's Republic of China

☯ These authors have contributed equally to this work.
* dingxixia@126.com (XD); redcellchen@163.com (YC); hzhou@smu.edu.cn (HZ); jfwchan@hku.hk (JFWC); 25760620@qq.com (WF)

## Abstract

Chikungunya fever (CF) is a mosquito-borne viral disease caused by Chikungunya virus (CHIKV) that is being increasingly reported in previously non-endemic areas, including Guangdong, China. Despite the low mortality rate associated with CF, some patients may develop chronic joint pain that persists for months or years. Moreover, due to the similarities in symptomatology between CHIKV and other arboviruses, clinical diagnosis without laboratory confirmation is inaccurate. The CHIKV envelope protein E1, critical for viral entry and assembly, is a promising diagnostic and therapeutic target. In this study, we expressed the CHIKV E1 protein in *Escherichia coli* and developed 16 monoclonal antibodies (mAbs) against E1 to establish an enzyme immunoassay (EIA). The EIA specifically detects CHIKV E1 with no cross-reactivity to other major human-pathogenic arboviruses, including dengue virus (DENV). We evaluated its performance in serum samples of 84 CHIKV-infected patients, with qRT-PCR as the reference standard, and compared it with 93 healthy and 94 DENV-1-infected controls. The assay achieved a sensitivity of 94.05% (95% CI: 87.3–97.3) and a specificity of 98.4% (95% CI: 95.3–99.5). This E1 antigen-based EIA enables rapid and accurate diagnosis of CHIKV infection to facilitate the control of outbreaks of this emerging arbovirus.

**Data availability statement:** All relevant data are within the manuscript and its Supporting Information files. Additional technical details are available upon reasonable request from the Research Board at Zhujiang Hospital via zjyykeyanchu@163.com.

**Funding:** This work was supported by the Guangdong Basic and Applied Basic Research Foundation (2023A1515220071 to XXD; 2022A1515011132 to YC; 2020A1515011171 to XXD) and the Partnership Programme of Enhancing Laboratory Surveillance and Investigation of Emerging Infectious Diseases and Antimicrobial Resistance for the Department of Health of the Hong Kong Special Administrative Region Government (to JFWC). The funders had no role in study design, data collection and analysis, decision to publish, or preparation of the manuscript.

**Competing interests:** The authors have declared that no competing interests exist.

## Author summary

A Critical Tool to Combat the Chikungunya Virus Threat: Climate change and urbanization are reshaping the geographical distribution of pathogens and their vectors. Among them, Chikungunya virus (CHIKV) is being increasingly detected in previously non-endemic areas and is becoming a major public health threat in tropical and subtropical areas. Rapid and accurate diagnosis is essential for controlling infectious disease outbreaks, but current methods like RT-PCR and traditional serological tests can be complex, time-consuming, and/or prone to cross-reactivity with other co-circulating arboviruses such as dengue virus. To address this urgent diagnostic need, the present study aimed to develop monoclonal antibodies (mAbs) targeting the CHIKV E1 protein. Using these antibodies, we successfully created a novel double-antibody sandwich enzyme immunoassay (EIA) with high sensitivity (94.05%) and specificity (98.40%). Importantly, the EIA did not show cross-reactivity with other major human-pathogenic arboviruses, making it a useful tool to complement existing diagnostics for CHIKV infection. Furthermore, these high-affinity mAbs are an essential foundation for creating rapid point-of-care (POC) tests to improve field surveillance and infection control efforts, especially in resource-limited settings where speed and simplicity are of paramount importance.

## Introduction

Chikungunya fever (CF), caused by the Chikungunya virus (CHIKV), is a mosquito-borne viral disease with ongoing global outbreaks. As of May 2025, approximately 220,000 cases and 80 CF-related deaths have been reported across the Americas, Africa, Europe, and Asia, including a recent imported outbreak in Shunde District of Foshan, Guangdong, China. CHIKV, an RNA virus of the Togaviridae family, is primarily transmitted by Aedes mosquitoes (Aedes aegypti and Aedes albopictus), with rare vertical [1] or blood-borne transmission [2]. Over 75% of infected individuals develop acute symptoms, including fever, rash, headache, and severe joint and muscle pain [3]. Polyarthralgia, the hallmark symptom, affects 87–98% of cases, while myalgia occurs in 46–59% [4]. The clinical overlap with other arboviral infections, such as dengue and Zika, often leads to misdiagnosis, delaying treatment and complicating management due to differing therapeutic needs (e.g., NSAID contraindications in dengue) [5]. Thus, accurate CHIKV diagnostics are critical to reduce misdiagnosis and optimize clinical outcomes.

The World Health Organization (WHO) advocates three primary laboratory tests for the diagnosis of CHIKV infection: virus isolation, serological testing, and polymerase chain reaction (PCR) [6]. Diagnostic method selection depends on the timing of sample collection relative to symptom onset. Virus isolation, although the gold standard, is time-consuming (≥7 days) and requires biosafety level 3 facilities, limiting its practical use. qRT-PCR offers high sensitivity and specificity but is less effective

beyond the first five days of infection due to a narrow detection window. Conversely, serological antibody testing achieves 100% positivity after five days but lacks sensitivity during the initial stages of infection (within five days of symptom onset) [7,8]. The integration of these two approaches can yield extensive coverage throughout the full duration of infection. The dependence on specialized equipment and skilled individuals for qRT-PCR testing constrains its applicability in expedited clinical or field environments. Therefore, there is an urgent need for development of straightforward, reliable, and rapid point-of-care testing (POCT) techniques to facilitate the comprehensive identification of CHIKV infection.

In comparison to nucleic acid testing (NAT), viral antigen detection offers a simple approach for the direct confirmation of CHIKV infection in individuals. The efficacy of antigen detection is significantly contingent upon high-quality monoclonal antibodies (mAbs) that can precisely identify specific viral antigens. Currently, no commercial CHIKV antigen detection assays are available. This study examined recent developments in CHIKV antigen detection technologies and identified the CHIKV-E1 protein as a viable target. The E1 protein sequence is notably conserved, minimizing the likelihood of mutation, and is secreted in substantial amounts during the acute viremic phase of infection [9]. The attributes of the E1 protein render it an optimal option for CHIKV antigen detection. Consequently, we produced monoclonal antibodies through recombinant E1 protein expression and animal immunization, enabling a double-antibody sandwich enzyme immunoassay (EIA). This assay demonstrates significant specificity for CHIKV, exhibits no cross-reactivity with other mosquito-borne viruses, and is well-suited for clinical specimen detection.

## Materials and methods

### Ethics statement

The study was approved by the ethics committee of Zhujiang Hospital, Southern Medical University (approval number: 2019-KY-081–01). All participants provided written informed consent for specimen storage. All experiments were performed in accordance with the Declaration of Helsinki and institutional guidelines.

The BALB/c mice (4–6 weeks old) were obtained from the center for Comparative Medicine Research (CCMR), the University of Hong Kong. The mice were housed in individually ventilated cages at an ambient temperature of 21–23°C, <65% humidity, with a 12-hour light-dark cycle. Animal use and experimental procedures followed protocols approved by the Committee on the Use of Live Animals in Teaching and Research, The University of Hong Kong (CULATR 24–456).

### Viruses, cells, and recombinant proteins

Chikungunya virus (CHIKV, Asian genotype), Yellow fever virus (YFV, 17D strain), Japanese encephalitis virus (JEV, SA14-14-2 strain), and the four serotypes of dengue virus (DENV-1, Hawaii strain; DENV-2, New Guinea-C strain; DENV-3, Guanxi-80–2 strain; DENV-4, H241 strain) were obtained from the Guangzhou Center for Disease Control and Prevention in China as reference strain. Cell culture supernatants infected with lentiviral-based pseudoviruses expressing the E1 protein of Getah virus (GETV) or Sindbis virus (SINV) were provided by the Department of Microbiology, The University of Hong Kong, for cross-reactivity testing. Recombinant proteins for specificity testing, including Mayaro virus (MAYV, Brazil strain) E1, SINV (AR339 strain) E1, Western Equine Encephalomyelitis Virus (WEEV, BFS1703 strain) E1, and Eastern Equine Encephalitis Virus (EEEV, 82V-2137 strain) E1, were purchased from Wuhan Yipu, China. Recombinant DENV envelope domain III (rDENV-ED3) protein was expressed in our laboratory using Pichia pastoris yeast and purified via Ni-nitrilotriacetic acid (Ni-NTA) affinity chromatography (QIAGEN, Germany). A commercial mouse anti-His mAb (Sigma-Aldrich, USA) was used to confirm the molecular weight of rCHIKV-E1 in Western blot analysis. MAbs for specificity testing are described in Preparation and Characterization of Monoclonal Antibodies Against CHIKV-E1 Protein. Aedes albopictus clone C6/36 cells were purchased from American Type Culture Collection (ATCC, CRL-1660) and used for virus culture, NS-1 myeloma cells (ATCC, TIB-18) was used for hybridoma fusion. All cell lines were confirmed mycoplasma-free.

## Virus culture

C6/36 cells were cultured in Minimal Essential Medium (MEM) supplemented with 10% fetal bovine serum (FBS) (ThermoFisher Scientific Inc Cat# 16140071) at 28°C in a humidified 5% $CO_2$ incubator and maintained until 80% confluence. For viral infection, confluent C6/36 monolayers were inoculated with viral stock at a multiplicity of infection (MOI) of 0.1. After 1 hour of adsorption at 28°C, the inoculum was removed, and cells were washed twice with phosphate-buffered saline (PBS). Then, the medium was replaced with MEM containing 2% FBS, and cultures were incubated under the same conditions. Virus-infected cells and supernatants were harvested when ~80% of cells exhibited cytopathic effects (CPE). Supernatants were clarified by centrifugation (3,000 × g, 10 min, 4°C) and stored at −80°C. Cell pellets were processed for further analysis.

## Preparation of CHIKV E1 recombinant protein

The E1 gene sequence of CHIKV was obtained from the Asian genotype virus strain CSB04010 (GenBank Accession No: KT175541) and West African (WA) genotype virus strain IbH35 (Genbank Accession No. HM045786). The full-length gene encoding the CHIKV E1 was synthesized by Shanghai Biotechnology Corporation (Shanghai, China) and cloned into the pET-21b (+) *Escherichia coli* (*E. coli*) expression vector. The recombinant plasmid was transformed into *E. coli* Origami B (DE3) cells for protein expression. Expression was induced with isopropyl β-D-1-thiogalactopyranoside (IPTG) under optimized conditions per manufacturer's guidelines. The recombinant CHIKV-E1 （rCHIKV-E1） protein was purified using Ni-nitrilotriacetic acid affinity chromatography (Ni-NTA) following the manufacturer's guidelines (QIAGEN, Germany). rCHIKV-E1 purity and integrity were assessed by sodium dodecyl sulfate-polyacrylamide gel electrophoresis and Western blotting with a mouse anti-His tag antibody.

## Preparation and characterization of monoclonal antibodies against CHIKV-E1 protein

The preparation and characterization of monoclonal antibodies (mAbs) against the rCHIKV-E1 protein were performed following established protocols [10,11]. Briefly, 4–6 week-old female BALB/c mice were subcutaneously injected with 50μg of rCHIKV-E1 protein emulsified in complete Freund's adjuvant (Sigma-Aldrich, Cat#: F5881). Booster immunizations were performed every 10 days using 30 μg of rCHIKV-E1 protein emulsified in incomplete Freund's adjuvant. After a total of 4 booster immunizations, serum samples were collected from the tail vein of mice and anti-rCHIKV-E1 antibody titers were measured by indirect EIA, using rCHIKV-E1 as the coated antigen. The mouse with the highest serum antibody titer was intraperitoneally injected with 100 μg of rCHIKV-E1 protein without adjuvant 3 days before cell fusion. Spleen cells from the immunized mouse were fused with NS-1 myeloma cells using polyethylene glycol (PEG) (Sigma-Aldrich, Cat#: P7181) to generate hybridomas. Hybridoma cells were screened by indirect EIA using rCHIKV-E1 (1μg/mL, 100μL/well) as the coated antigen and the supernatant of positive cells was further identified by immunofluorescence (IFA) and Western blotting assay (WB). The subtypes of the monoclonal antibodies were determined using a commercial mouse monoclonal antibody subtyping kit (Zymed Laboratories, Carlsbad, CA). Finally, specific monoclonal antibodies were purified from ascites fluid via protein G chromatography (Biodragon, China) according to the manufacturer's instructions. The purified mAbs were conjugated with horseradish peroxidase (HRP; Sigma-Aldrich, USA) using periodate oxidation. Additionally, mAbs against DENV-ED3 (5A1A3, 5F29A6), DENV-NS1 (5A11A5), and WNV-NS1 (WNVNS1-M66, WNVNS1-M2), used as negative controls for specificity testing (Fig 1B), were produced in our laboratory using similar immunization and hybridoma protocols.

## Analysis of the antibody-binding kinetics

The binding affinity of the mAbs to the rCHIKV-E1 protein was determined using Biological Layer Interferometry (BLI) on the Gator system (GatorBio, Suzhou, China). The anti-mouse IgG Fc biosensors were equilibrated in Q buffer and used to

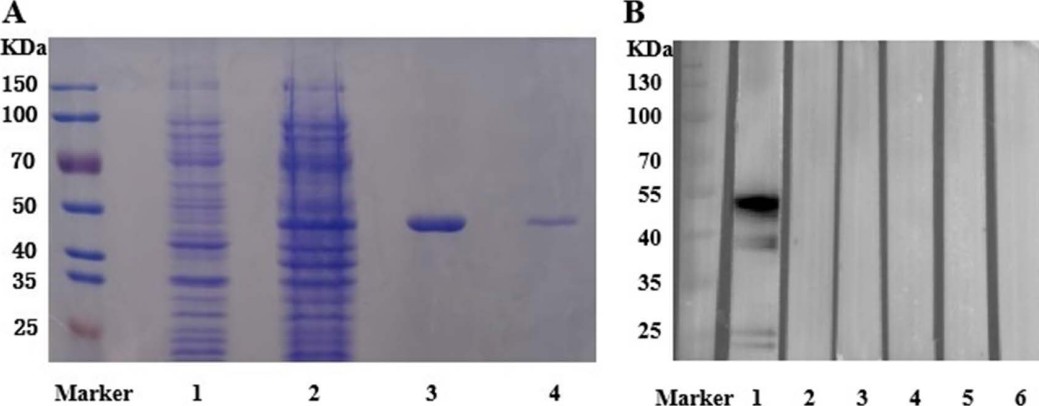

**Fig 1. Analysis of rCHIKV-E1 Protein Expression and Purification. (A)** Sodium dodecyl sulphate-polyacrylamide gel electrophoresis (SDS-PAGE) showing purity of His-tagged E1 protein of CHIKV. Marker, prestained protein ladder; Lane 1, Without IPTG induced crude *E. coli* lysates; Lane 2, IPTG-induced E. coli lysate expressing pET-21b (+) with CHIKV-E1 cDNA; Lane 3, purified rCHIKV-E1 protein; Lane 4, refolded rCHIKV-E1 protein. **(B)** Western blot analysis of refolded rCHIKV-E1 to confirm molecular weight and specificity using monoclonal antibodies (mAbs) produced in our laboratory. Lane 1: mouse anti-His mAb (molecular weight confirmation); Lanes 2–3: mouse anti-DENV-ED3 mAbs (5A1A3, 5F29A6, negative controls); Lane 4: mouse anti-DENV-NS1 mAb (5A11A5, negative control); Lanes 5–6: mouse anti-WNV-NS1 mAbs (WNVNS1-M66, WNVNS1-M2, negative controls).

capture 10µg/mL of mAb for 240 seconds, after equilibrating in Q buffer for 60 seconds, the sensors were then combined with serially diluted rCHIKV-E1 protein (experimental group) or Q buffer (negative control group) for 180 seconds, followed by dissociated in Q buffer for 300 seconds. Gator System Analysis software was used to calculate association (Ka) and dissociation (Kd) kinetic rate constants, and equilibrium dissociation constant (KD) of these mAbs.

## Epitope identification of monoclonal antibodies

A competitive EIA was conducted utilizing the rCHIKV-E1 protein as the coated antigen to identify the binding epitopes of isolated mAbs. The detailed experimental procedure and results are described below: The rCHIKV-E1 protein (1 µg/mL, 100µl/well) was coated onto a 96-well plate (Costar Corning, USA) and incubated overnight at 4°C. The plate was then washed and blocked with a blocking buffer (0.25% casein-PBS) to prevent non-specific binding. Unlabeled mAb (500 µg/mL) was incubated with HRP-labeled mAb (1:1000) for 1 h at 37°C. After the plates were washed, the binding of the HRP-labelled mAb was detected by the addition of tetramethylbenzidine (TMB) substrate (KPL, Gaithersburg, MD). The reaction was stopped after 10 min by the addition of 0.3 N sulfuric acid, and the plates were then examined in an EIA plate reader (Bio-Tek, Winooski, VT). An unlabeled mAb specific for influenza nonstructural protein 1 was used as an irrelevant control [12]. The distinct binding epitopes of mAbs were determined as described previously [13].

## Establishment of the double-antibody sandwich EIA for CHIKV-E1 protein detection

A double-antibody sandwich EIA was constructed to establish a sensitive and specific antigen detection approach for CHIKV, using an optimized combination of capture and detection antibodies [11]. In brief, A 96-well microplate (Costar, Corning) was coated with 100 µL/well of the capture antibody at an optimized concentration (15 µg/mL). The plate was incubated overnight at 4°C to allow the capture antibody to bind to the plate. After coating, the plate was washed to remove unbound antibodies. The wells were blocked with 0.25% casein-PBS blocking solution (260 µL/well) and incubated at 4°C for 24 hours to prevent nonspecific binding. Serial dilutions of the rCHIKV-E1 protein (100 µL/well) were added to the wells, and rDENV-ED3 protein was used as a negative control to ensure specificity. The plate was incubated at 37°C for 1 hour for antigen binding. After washing the plate to remove unbound antigen, 100 µL/well of HRP-labelled monoclonal antibody at its working concentration

was added and incubated for 30 min at 37°C. After the plates were washed, and the binding of the HRP-labelled mAbs were detected by the addition of TMB substrate (KPL, Gaithersburg, MD). The reaction was stopped by adding 0.3 N sulfuric acid (100 μL/well), and the plates then were examined in an EIA plate reader (Bio-Tek, Winooski, VT).

### Sensitivity and specificity of the detection method

To evaluate the sensitivity and specificity of the double-antibody sandwich EIA for CHIKV antigen detection, experiments were conducted using various arboviruses and recombinant proteins as our previous study [10]. Formalin-inactivated virus supernatants of CHIKV, DENV-1, DENV-2, DENV-3, DENV-4, JEV, YFV, ZIKV, GETV, SINV were used in this study. Virus supernatants were serially diluted (2-fold, starting at 1:10, total 16 dilutions). The EIA detected E1 protein in virus supernatants, with MEM medium as a negative control. RCHIKV-E1 protein was serially diluted (2-fold, starting at 1 μg/mL), with rDENV-ED3, rMAYV-E1, rSINV-E1, rWEEV-E1, and rEEEV-E1 protein as negative controls to assess cross-reactivity.

### Clinical specimens

Serum samples were collected from 84 patients with confirmed Chikungunya virus (CHIKV) infection (diagnosed by qRT-PCR) during the recent imported CHIKV outbreak in Shunde District of Foshan, Guangdong, China. Additionally, 94 serum samples from patients with confirmed dengue virus serotype 1 (DENV-1) infection (from the 2014 Guangzhou epidemic, qRT-PCR-confirmed) and 93 serum samples from healthy volunteers (archived in 2019) were included. All samples were leftover clinical specimens obtained from Zhujiang Hospital, Southern Medical University, Guangzhou, China. Blood samples were collected in plain tubes, centrifuged at 3,000 × g for 10 min at 4°C to isolate serum, and stored at −80°C until analysis. All participants provided written informed consent for specimen storage and use, as approved by the Ethics Committee of Zhujiang Hospital (approval number: 2019-KY-081–01).

### Sample preparation and testing

The CHIKV-E1 enzyme immunoassay (EIA) was tested on mouse and human serum samples. For mouse samples, fifteen 3-day-old suckling mice were intracranially injected with 20 μL of CHIKV viral culture media ($1.2 \times 10^4$ PFU/mouse). By day 3 post-infection, six mice died, while others exhibited motor retardation and anorexia. Surviving mice were euthanized, and blood samples were collected from the carotid artery, allowed to clot at room temperature, centrifuged at 3,000 × g for 10 min at 4°C to isolate serum. Total viral RNA was extracted from the remaining blood clots immediately using RNAiso Plus (TaKaRa, China). The double-antibody sandwich EIA detected CHIKV-E1 antigen in nine mouse serum samples, with qRT-PCR confirming CHIKV RNA (using non-infection mouse serum as a negative control). Human serum samples (described in Clinical Specimens) from 84 CHIKV-infected patients, 94 DENV-1-infected patients, and 93 healthy controls were tested to evaluate EIA sensitivity and specificity.

### Statistical analysis

OD distributions between patients and controls were compared using the Wilcoxon rank-sum test. A receiver-operator characteristic (ROC) curve was produced, illustrating all sensitivity and specificity pairs for various OD cut-off points utilizing Prism 10.5.0. The discriminatory ability between cases and controls was assessed by computing the area under the ROC curve (AUC) along with the 95% confidence intervals (CIs). CHIKV-infected serum samples were analyzed by two-way ANOVA, with $P < 0.05$ considered statistically significant.

## Results

### CHIKV-E1 protein expression and preparation of monoclonal antibodies against E1 protein

The rCHIKV E1 protein was produced in *E. coli* using the pET-21b (+) vector. The protein, produced as inclusion bodies, was subjected to washing to eliminate contaminants, denatured with urea, purified with a Ni-nitrilotriacetic acid (Ni-NTA)

affinity chromatography kit (QIAGEN), and refolded through stepwise dialysis against 1 × PBS to reinstate its native structure. We utilized non-induced crude cell lysates of the transformed CHIKV-E1 construct, induced crude cell lysate, purified rCHIKV-E1 protein, and refolded rCHIKV-E1 protein for sodium dodecyl sulfate polyacrylamide gel electrophoresis (SDS-PAGE) analysis. The results indicate that following induction with IPTG, purification, and refolding, a distinct ~45 kDa band, consistent with the expected 47.42 kDa molecular weight, was observed post-induction, purification, and refolding, absent in non-induced lysates (Fig 1A). Western blot analysis confirmed the molecular weight using a mouse anti-His tag mAb and demonstrated high specificity of rCHIKV-E1, with no cross-reactivity to mAbs against DENV-ED3 (5A1A3, 5F29A6), DENV-NS1 (5A11A5), or WNV-NS1 (WNVNS1-M66, WNVNS1-M2). (Fig 1B). Immunofluorescence assays using mouse anti-serum confirmed specific binding of rCHIKV-E1 to CHIKV-infected C6/36 cells (S1 Fig).

## Monoclonal antibody (mAb) preparation and identification

BALB/c mice were immunized with rCHIKV-E1 four times at 10-day intervals. Splenocytes from immunized mice were fused with NS-1 myeloma cells and selected in hypoxanthine-aminopterin-thymidine (HAT) medium. Hybridoma supernatants were screened by EIA using rCHIKV-E1 as the antigen, and clones exhibiting high reactivity ($OD_{450-630} > 1.0$) were selected for further characterization. Chosen monoclonal antibodies were evaluated using immunofluorescence against CHIKV-infected C6/36 cell smears and DENV (serotypes 1–4), JEV, or YFV. Furthermore, we employed Western blot analysis to verify their binding to CHIKV-E1. All 16 monoclonal antibodies generated displayed robust reactivity to CHIKV-infected C6/36 cells, with 13 exhibiting high specificity for CHIKV and no cross-reactivity with related flaviviruses. M16 showed cross-reactivity with DENV1–4; M9 exhibited strong cross-reactivity with DENV 1, 3, and 4; M15 demonstrated cross-reactivity with DENV 1 and 4, while none exhibited cross-reactivity with YFV and JEV (Table 1). The mAbs comprised eight IgG2a and eight IgG1.

Table 1. Immunological characteristics of anti-CHIKV-E1 mouse monoclonal antibodies.

| Hybridoma clone no. | Isotype | Epitope group | Kinetics | EIA | IFA | | | | | | | WB |
|---|---|---|---|---|---|---|---|---|---|---|---|---|
| | | | | rCHIKV-E1 | CHIKV | DENV-1 | DENV-2 | DENV-3 | DENV-4 | YFV | JEV | rCHIKV-E1 |
| CH-M2 | IgG1 | I | 1.60E-06 | 2.229 | ++ | – | – | – | – | – | – | ± |
| CH-M3 | IgG2a | II | 2.88E-10 | 2.245 | + | – | – | – | – | – | – | + |
| CH-M4 | IgG1 | VI | 1.00E-12 | 2.448 | + | – | – | – | – | – | – | + |
| CH-M5 | IgG1 | I | 7.77E-09 | 1.899 | +-++ | – | – | – | – | – | – | – |
| CH-M6 | IgG1 | VII | 7.04E-10 | 2.219 | ++ | – | – | – | – | – | – | ND |
| CH-M7 | IgG2a | VIII | 1.58E-10 | 2.057 | ++ | – | – | – | – | – | – | ND |
| CH-M8 | IgG2a | IX | 6.25E-10 | 2.164 | ++ | – | – | – | – | – | – | + |
| CH-M9 | IgG1 | I | 1.92E-09 | 2.268 | + | +++ | – | +++ | +++ | – | – | + |
| CH-M10 | IgG1 | IV | 1.08E-09 | 2.131 | ± | – | – | – | – | – | – | + |
| CH-M11 | IgG1 | IV | 4.40E-09 | 1.864 | ± | – | – | – | – | – | – | ± |
| CH-M14 | IgG1 | IV | 9.51E-10 | 2.019 | ± | – | – | – | – | – | – | + |
| CH-M15 | IgG2a | V | 1.29E-06 | 1.589 | ± | ± | – | – | ± | – | – | + |
| CH-M16 | IgG2a | II | 6.12E-09 | 2.584 | ++ | + | ± | + | + | – | – | + |
| CH-M17 | IgG2a | II | 4.97E-10 | 1.844 | ± | – | – | – | – | – | – | + |
| CH-M18 | IgG2a | III | 7.66E-10 | 2.658 | ++ | – | – | – | – | – | – | + |
| CH-M19 | IgG2a | III | 7.57E-09 | 2.791 | + | – | – | – | – | – | – | + |

Note: EIA, Enzyme immunoassay; IFA, Immunofluorescence Assay; WB, Western blot; CHIKV, Chikungunya virus; DENV-1, Dengue virus 1; DENV-2, Dengue virus 2; DENV-3, Dengue virus 3; DENV-4, Dengue virus 4; YFV, Yellow fever virus; JEV, Japanese encephalitis virus; ND, Not detected.

## Analysis of binding kinetics and binding epitope of monoclonal antibodies

The binding kinetics of each E1-binding mAbs were assessed using biolayer interferometry (BLI). The mAbs exhibited Kd values ranging from <1 pM to 1.6µM. (Fig 2A). Subsequently, we identified the binding epitopes of mAbs by a competitive EIA involving 16 E1-binding mAbs. MAbs with >70% mutual blocking were grouped into the same epitope category. We categorized our 16 E1-binding monoclonal antibodies into 9 groups based on the competitive inhibition patterns of mAb pairings (Fig 2B).

## Development and assessment of a double-antibody sandwich EIA

A double-antibody sandwich EIA was established using monoclonal antibodies targeting distinct CHIKV-E1 epitopes for sensitive and specific antigen detection. Among the nine high-sensitivity antibody pairs found, the combination of CH-M11 (capture) and CH-M6 (detection) was deemed the most effective based on performance metrics. The assay was refined using a coating concentration of 15 µg/mL for CH-M11, a 1:500 dilution of HRP-labeled CH-M6, and incubation conditions of 1 hour at 37°C for the antigen and 40 minutes at 37°C for the detection antibody.

The sensitivity and specificity of the double-antibody sandwich EIA for detecting CHIKV-E1 were evaluated using serial dilutions of recombinant proteins, rCHIKV-E1, rMAYV-E1, rSINV-E1, rWEEV-E1, rEEEV-E1, and rDENV-ED3 to ascertain the limit of detection (LOD). The test achieved a detection limit of 122 pg/mL for rCHIKV-E1. No cross-reactivity was detected with rMAYV-E1, rSINV-E1, rWEEV-E1, rEEEV-E1, or rDENV-ED3 protein, hence demonstrating specificity for CHIKV-E1 (Fig 3A). Subsequently, we further assessed the EIA performance using formalin-inactivated culture supernatants from CHIKV, DENV, JEV, YFV, ZIKV, GETV, and SINV. The test confirmed a detection limit of 4.8 PFU/mL for CHIKV (1:40960 dilution) and no cross-reactivity with other viruses, demonstrating high sensitivity for CHIKV (Fig 3B).

## Cut-off value setting and sample testing

To evaluate the CHIKV-E1 EIA's sensitivity, serum samples from nine CHIKV-infected suckling mice were analyzed. CHIKV RNA was detected in all samples by qRT-PCR (Ct value range: 15.18-29.77). Fig 4A shows significant OD distribution differences between animal cases and controls ($P < 0.0001$), with similar differences for YFV- and DENV-1-infected mouse serum ($P < 0.0001$). Fig 4B shows the ROC curve plotting true positives (sensitivity) against false positives (specificity). The AUC was 1.0, indicating perfect discriminatory accuracy in the animal model.

To establish a reliable cutoff value for the double-antibody sandwich enzyme immunoassay (EIA) targeting the Chikungunya virus (CHIKV) E1 antigen, optical density (OD) values at 450 and 630 nm were measured in sera from 93 healthy donors. The mean $OD_{450-630}$ was 0.0254 (SD 0.0035), resulting in a cutoff value of 0.0359 (mean + 3 × SD). Of the 93 healthy control samples, 91 were below this cutoff (true negatives), and 2 were false positives. The assay was further evaluated using 84 sera from CHIKV-infected patients (confirmed by qRT-PCR) collected during the recent imported CHIKV outbreak in Shunde District of Foshan, Guangdong, China, and 94 sera from patients infected with Dengue virus serotype 1 (DENV-1; Guangzhou, 2014 epidemic, validated by qRT-PCR). The CHIKV E1 EIA correctly identified 79 of 84 Chikungunya cases as positive (sensitivity: 94.05%, 95% CI: 87.3-97.3) and 91 of 93 healthy controls as negative (specificity: 97.85%, 95% CI: 93.5-99.4). Among the 94 DENV-1 samples, 93 were negative, with 1 false positive (specificity: 98.94%, 95% CI: 94.3–99.8). Combining healthy and DENV-1 controls (n = 187), the overall specificity was 98.40% (95% CI: 95.3–99.5). Table 2 summarizes the diagnostic performance metrics.

A significant difference in $OD_{450-630}$ values was observed between Chikungunya cases and healthy controls (Mann-Whitney U test, $p < 0.0001$), as well as between Chikungunya and DENV-1 cases (Mann-Whitney U test, $p < 0.0001$). Fig 4C illustrates the distribution of $OD_{450-630}$ values across Chikungunya cases, healthy controls, and DENV-1 cases, demonstrating the assay's ability to distinguish CHIKV from healthy and DENV-1-infected individuals. The area under the receiver operating characteristic (ROC) curve (AUC) was 0.9743 (95% CI: 0.9462–1), indicating excellent discriminatory performance (Fig 4D).

PLOS Neglected Tropical Diseases

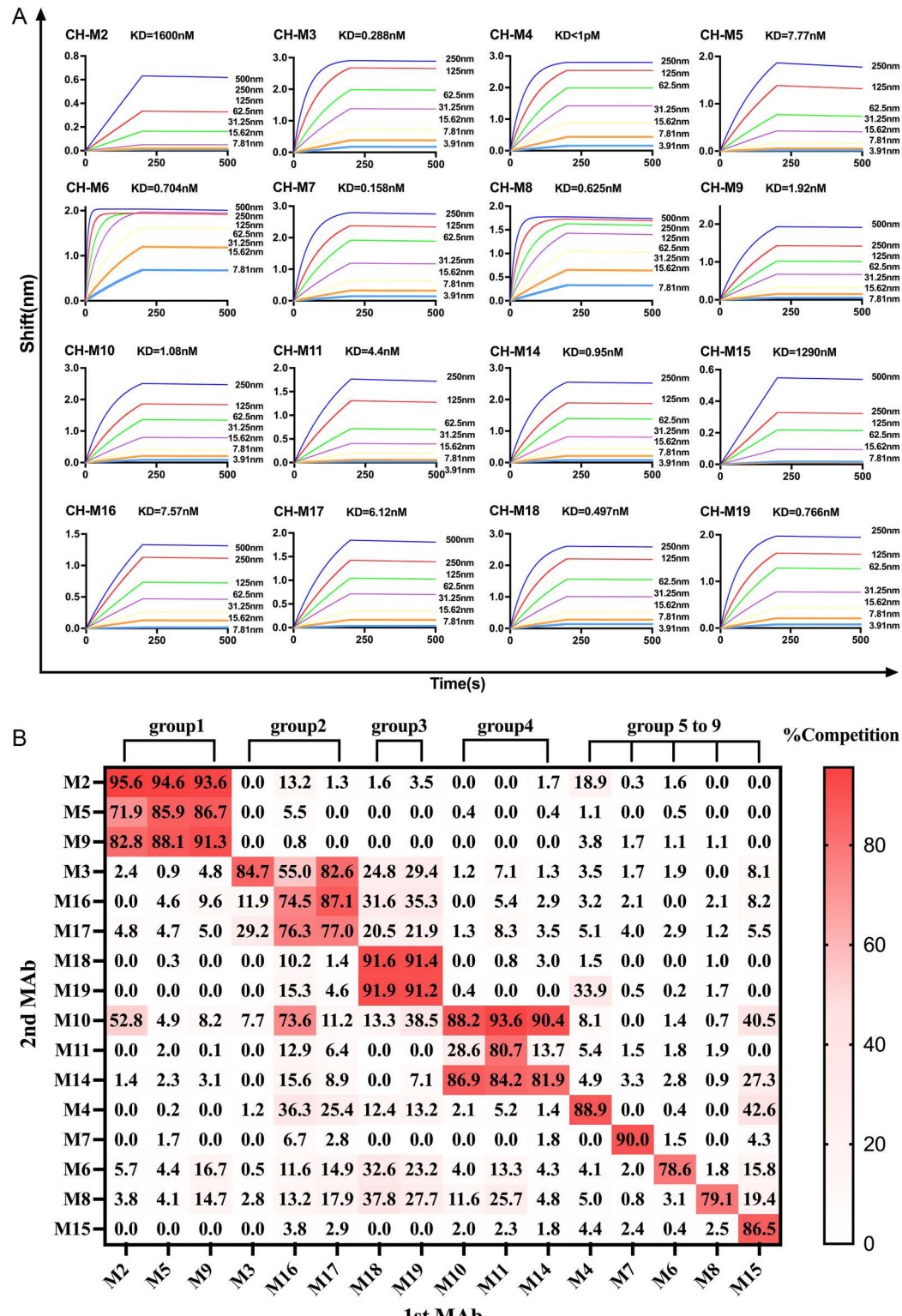

**Fig 2. Binding Kinetics and Epitope Mapping of Anti-CHIKV-E1 mAbs. (A)** Binding kinetics of selected mouse anti-E1 mAbs with the recombinant CHIKV E1 protein were measured by biolayer interferometry. Anti-mouse IgG Fc probes with immobilized antibody (10μg/mL) were exposed to serial dilutions of rCHIKV-E1 (1pM to 1.6μM). **(B)** Heat map showing competitive inhibition (%) of 16 anti-CHIKV-E1 mAbs, mAbs were categorized into nine

groups, with each group exhibiting reactivity to identical or sterically overlapping epitopes on the CHIKV E1 antigens. Competitive inhibition values exceeding 75% are designated as the same epitopes.

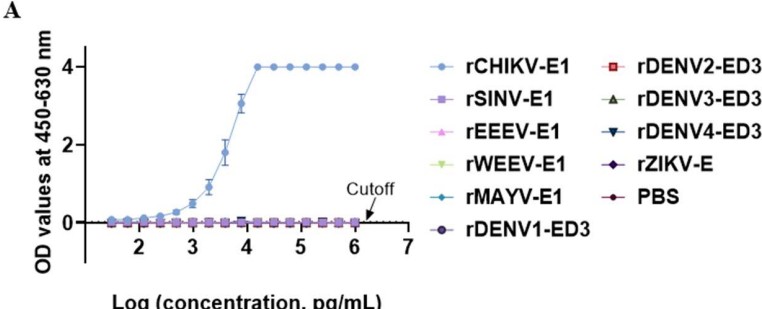

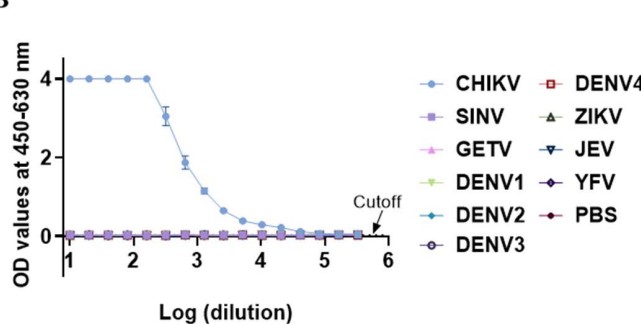

**Fig 3. Detection limit and cross reactivity of the double-antibody sandwich EIA. (A)** Detection limit of rCHIKV-E1 (Asian-genotype strain) using the double-antibody sandwich EIA, with mAb M11 (capture) and mAb M6 (detection). The EIA evaluated recombinant E1 proteins from CHIKV, Mayaro virus (MAYV), Western equine encephalitis virus (WEEV), Sindbis virus (SINV), Eastern equine encephalitis virus (EEEV), and Zika virus (ZIKV). Recombinant envelope domain III protein from four serotypes of dengue virus (rDENV-ED3) was also included. **(B)** Detection of authentic CHIKV in culture supernatants, tested for cross-reactivity with DENV, Japanese encephalitis virus (JEV), yellow fever virus (YFV), Zika virus (ZIKV), and cell culture supernatants infected with lentiviral-based pseudoviruses expressing Getah virus (GETV) or SINV E1 protein, using MEM medium as a negative control. Dashed lines represent the cutoff values (mean + 3 × SD based on $OD_{450-630}$ values devoid of antigens) for each EIA.

## Discussion

Global warming, migration, and urbanization have accelerated chikungunya virus (CHIKV) spread, posing a global health challenge, particularly in regions like Shunde District of Foshan, Guangdong, China, where a recent imported outbreak occurred. Chikungunya fever, characterized by fever, rash, severe joint pain, and chronic arthritis, impairs quality of life. Controlling CHIKV requires antiviral drugs, vaccines, and rapid, reliable diagnostics for early detection and surveillance. Current diagnostics-virus isolation, RT-PCR, and serological testing-have limitations. RT-PCR is highly sensitive but complex and impractical in resource-limited settings due to specialized equipment and expertise requirements. Serological tests are simple but prone to cross-reactivity with other arboviruses (e.g., dengue, Zika, Sindbis) and ineffective in early infection, unable to distinguish recent from past infections. These shortcomings underscore the need for antigen detection methods, which enable early diagnosis by targeting viral proteins in the acute phase, complementing or replacing RT-PCR [14]. Antigen detection methods offer the advantage of early diagnosis by directly detecting viral proteins during the acute phase of infection. Although antigen detection methods targeting CHIKV E1/E2 proteins have been reported [9,14–18], they lack commercial availability and require further clinical validation.

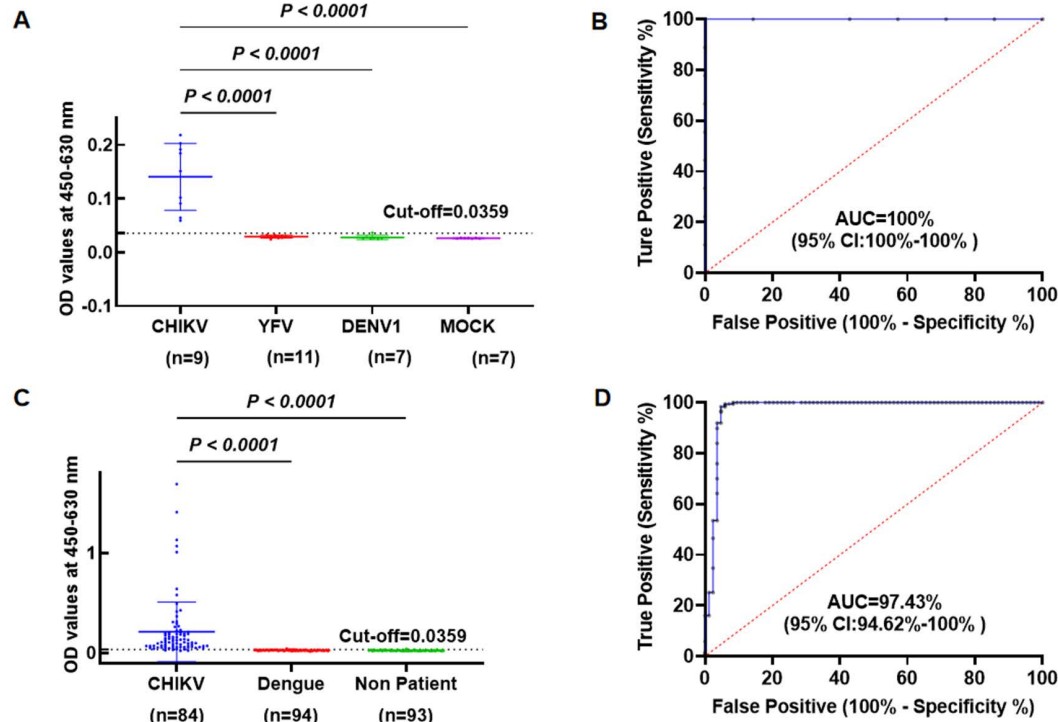

**Fig 4. Efficacy of the double-antibody sandwich EIA in CHIKV diagnosis. (A)** Sensitivity and specificity of the EIA for CHIKV-infected (CHICK: n = 9), YFV-infected (YFV: n = 11), DENV-1-infected (DENV1: n = 7), and PBS-inoculated (MOCK: n = 7, control) suckling mouse sera, diluted 1:10, showing $OD_{450-630}$ values. **(B)** Receiver Operating Characteristic (ROC) curve showing an AUC of 1.0, indicating perfect discrimination between CHIKV-infected and control (YFV, DENV-1, PBS) mouse sera. **(C)** OD450–630 distribution of CHIKV-infected human sera (CHIKV: n = 84, from the outbreak in Shunde, Guangdong), healthy controls (Non patient: n = 93), and DENV-1-infected sera (DENV: n = 94), with significant differences (P < 0.0001). Dashed lines represent the cut-off values (mean + 3 × SD based on OD450–630 values of healthy controls). **(D)** ROC curve exhibited exceptional discrimination, achieving 97.43% accuracy, between CHIKV-infected cases and non-CHIKV-infected (DENV and Healty) controls.

High-quality antigens are critical for generating specific mAbs. Soluble rCHIKV-E1 was produced by optimizing inclusion body renaturation, retaining native structure and immunogenicity, as evidenced by specific immune responses in mice. Based on this antigen, a panel of mAbs specific for the CHIKV-E1 protein was developed. To ensure mAbs recognized all three CHIKV genotypes (ECSA, West African, Asian) and minimized missed diagnoses, mice were immunized with Asian-genotype rCHIKV-E1, and resulting mAbs were screened for cross-reactivity with ECSA-genotype CHIKV-infected cells (IFA) and WA-genotype rCHIKV-E1 (EIA), yielding high-affinity mAbs effective across genotypes.

Using these mAbs, a double-antibody sandwich EIA was developed with high sensitivity (LOD: 4.8 PFU/mL for CHIKV supernatant; 122 pg/mL for rCHIKV-E1) and specificity, showing no cross-reactivity with arboviruses such as dengue, Japanese encephalitis, yellow fever, Zika, Getah, and Sindbis viruses. Sera from 9 CHIKV-infected suckling mice (all RNA-positive) tested 100% positive for CHIKV-E1 antigen and did not exhibit cross-reactivity with YFV and DENV-1. A cutoff value of 0.0359 was established based on the mean $OD_{450-630}$ of 93 healthy control sera. The assay achieved a sensitivity of 94.05% in 84 sera from confirmed Chikungunya patients and specificities of 97.85% and 98.94% in 93 healthy and 94 DENV-1 sera, respectively (combined specificity: 98.40%, 95% CI: 95.3–99.5). The area under the ROC curve (AUC) was 0.9743 (95% CI: 0.9462–1), indicating excellent discriminatory performance.

**Table 2. Diagnostic performance of the enzyme immunoassay (EIA) for detecting E1 antigen in sera from 84 confirmed Chikungunya patients, 93 healthy controls, and 94 DENV-1-infected patients.**

| Parameter | Value (95% CI) |
|---|---|
| **Raw Counts** | |
| True Positive (TP) | 79 |
| False Negatives (FN) | 5 |
| True Negatives (TN, Healthy) | 91 |
| False Positives (FP, Healthy) | 2 |
| True Negatives (TN, DENV-1) | 93 |
| False Positives (FP, DENV-1) | 1 |
| **Performance Metrics** | |
| Sensitivity (%) | 94.05 (87.3–97.3) |
| Specificity (Healthy, %) | 97.85 (93.5–99.4) |
| Specificity (DENV-1, %) | 98.94 (94.3–99.8) |
| Specificity (Combined, %) | 98.40 (95.3–99.5) |
| Positive Likelihood Ratio | 43.74 |
| Negative Likelihood Ratio | 0.06 |

Note: The cutoff value for positivity was determined as the mean optical density (OD) of healthy control sera (0.0254±SD 0.0035) plus 3 standard deviations, resulting in a cutoff OD of 0.0359. CI, confidence interval.

This study developed a highly sensitive and specific CHIKV-E1 antigen sandwich EIA, ideal for early diagnosis and environmental mosquito monitoring, offering substantial clinical and public health potential. With further validation in diverse clinical settings and potential commercialization, this assay could enhance CHIKV prevention and control.

## Limitations of this study

The CHIKV-E1 sandwich EIA requires laboratory infrastructure for plate-based processing, limiting its use in resource-constrained settings. However, the high-affinity and specific mAbs developed in this study provide a foundation for simpler, rapid antigen detection methods, such as colloidal gold or latex particle-based lateral flow assays or fluorescence-based tests, which could enhance field applicability. The lack of CHIKV prevalence data in the study population precluded calculation of positive and negative predictive values, essential for assessing clinical utility across epidemiological settings. Additionally, the assay's performance in diverse epidemiological contexts beyond the Guangdong outbreak requires further evaluation. Future research requires international collaboration, larger sample sizes, and development of point-of-care detection methods for community-based infection management.

## Supporting information

**S1 Data. Raw data of Table 1.**
(XLSX)

**S2 Data. Raw data of Fig 2.**
(XLS)

**S3 Data. Raw data of Fig 3.**
(XLS)

**S4 Data. Raw data of Fig 4.**
(ZIP)

**S1 Fig. Immunofluorescence Assay (IFA) detection of anti-CHIKV-E1 antibodies in immunized mouse serum.**
(DOCX)

## Author contributions

**Conceptualization:** Kun Wen, xixia ding.

**Data curation:** Zhihong Zhou, Peipei Xu, Biao Di, Jialing Song, Cuilian Yang, Lili Zhan, Wenxi Feng, Yue Chen, xixia ding.

**Formal analysis:** Kun Wen, Biao Di, Shuofeng Yuan, xixia ding.

**Funding acquisition:** Jasper Fuk-Woo Chan, Yue Chen, xixia ding.

**Investigation:** Zhihong Zhou, Biao Di, Jialing Song, Jianpiao Cai, Shuofeng Yuan, Yue Chen, xixia ding.

**Methodology:** Peipei Xu, Mei Xu, xixia ding.

**Project administration:** xixia ding.

**Resources:** Biao Di, Chongquan Zhao, Wenxi Feng, xixia ding.

**Supervision:** Jasper Fuk-Woo Chan, xixia ding.

**Validation:** Wenxi Feng, Hongwei Zhou, xixia ding.

**Visualization:** Jianpiao Cai, xixia ding.

**Writing – original draft:** Kun Wen, xixia ding.

**Writing – review & editing:** Kun Wen, Kelvin Kai-Wang To, Wenxi Feng, Jasper Fuk-Woo Chan, Hongwei Zhou, xixia ding.

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
