## [Decision Letter · Decision Letter 0]

23 Sep 2025

Preparation of monoclonal antibodies specific for the E1 protein of chikungunya virus and the establishment of the antigen detection method

Dear Dr. ding,

Thank you for submitting your manuscript to PLOS Neglected Tropical Diseases. After careful consideration, we feel that it has merit but does not fully meet PLOS Neglected Tropical Diseases's publication criteria as it currently stands. Therefore, we invite you to submit a revised version of the manuscript that addresses the points raised during the review process.

Please submit your revised manuscript within 60 days Nov 22 2025 11:59PM. If you will need more time than this to complete your revisions, please reply to this message or contact the journal office at plosntds@plos.org. Please include the following items when submitting your revised manuscript:

We look forward to receiving your revised manuscript.

Kind regards,

Han Xia, Ph.D.

Academic Editor

Paul Brindley

Editor-in-Chief

Shaden Kamhawi

co-Editor-in-Chief

Paul Brindley

co-Editor-in-Chief

**Additional Editor Comments (if provided):**

Reviewer #1:

Reviewer #2:

**Journal Requirements:**

**Reviewers' Comments:**

Reviewer's Responses to Questions

**Key Review Criteria Required for Acceptance?**

**Methods**

-Are the objectives of the study clearly articulated with a clear testable hypothesis stated?

-Is the study design appropriate to address the stated objectives?

-Is the population clearly described and appropriate for the hypothesis being tested?

-Is the sample size sufficient to ensure adequate power to address the hypothesis being tested?

-Were correct statistical analysis used to support conclusions?

-Are there concerns about ethical or regulatory requirements being met?

Reviewer #1: See the general comments

Reviewer #2: All yes

**Results**

-Does the analysis presented match the analysis plan?

-Are the results clearly and completely presented?

-Are the figures (Tables, Images) of sufficient quality for clarity?

Reviewer #1: See the general comments

Reviewer #2: -Does the analysis presented match the analysis plan? Yes

-Are the results clearly and completely presented? Yes

-Are the figures (Tables, Images) of sufficient quality for clarity? From the pdf file that I received, the quality was far from excellent, please provide better ones

**Conclusions**

-Are the conclusions supported by the data presented?

-Are the limitations of analysis clearly described?

-Do the authors discuss how these data can be helpful to advance our understanding of the topic under study?

-Is public health relevance addressed?

Reviewer #1: The conclusion is not well supported on results.

Reviewer #2: All yes

**Editorial and Data Presentation Modifications?**

Reviewer #1: Major

Reviewer #2: Minor revision. Recently there are Chikv outbreaks in China and by now the authors must already have these human sera, I would recommend to have the additional validation analysis.

**Summary and General Comments**

Reviewer #1: Comments on the manuscript describing the establishment of CHIKV Antigen ELISA:

1. The main weakness of this study is that the system was not evaluated using human serum samples. Without such validation, it is difficult to trust its suitability for diagnostic purposes.

2. The authors should test at least a small panel (e.g., 10 human serum samples) to verify the reliability of the ELISA system.

3. Since the same recombinant CHIKV E1 antigen was used for mouse immunization to generate monoclonal antibody clones, the ELISA system may be adequate for mouse-derived samples. However, the authors must demonstrate its applicability using human CHIKV-infected serum samples.

4. Among the monoclonal antibodies generated, the authors should clarify the criteria used to select the antibody for HRP conjugation.

5. In Table 1, some clones are positive for ELISA and IFA but negative for WB. The authors need to explain this discrepancy.

6. The statement that there is no CHIKV outbreak in China is incorrect. In 2025, a CHIKV outbreak occurred in China. Please refer to the following sources:

https://sangerinstitute.blog/2025/08/07/chikungunya-virus-in-china/

https://www.nature.com/articles/d41586-025-02794-2

7. As CHIKV outbreaks have also been reported in countries neighboring China, the authors should consider collaborating with regional partners to validate their ELISA system using human serum samples from outbreak settings.

Reviewer #2: None applicable

PLOS authors have the option to publish the peer review history of their article (what does this mean? ). If published, this will include your full peer review and any attached files.

**Do you want your identity to be public for this peer review?** For information about this choice, including consent withdrawal, please see our Privacy Policy .

Reviewer #1: No

Reviewer #2: No

**Figure resubmission:**
---

## [Decision Letter · Decision Letter 1]

24 Nov 2025

Dear Professor ding,

We are pleased to inform you that your manuscript 'Development of monoclonal antibodies against E1 protein of Chikungunya virus' has been provisionally accepted for publication in PLOS Neglected Tropical Diseases.

Best regards,

Han Xia, Ph.D.

Academic Editor

Paul Brindley

Editor-in-Chief

Shaden Kamhawi

co-Editor-in-Chief

Paul Brindley

co-Editor-in-Chief

Reviewer's Responses to Questions

**Key Review Criteria Required for Acceptance?**

**Methods**

-Are the objectives of the study clearly articulated with a clear testable hypothesis stated?

-Is the study design appropriate to address the stated objectives?

-Is the population clearly described and appropriate for the hypothesis being tested?

-Is the sample size sufficient to ensure adequate power to address the hypothesis being tested?

-Were correct statistical analysis used to support conclusions?

-Are there concerns about ethical or regulatory requirements being met?

Reviewer #1: yes.

Reviewer #2: (No Response)

**Results**

-Does the analysis presented match the analysis plan?

-Are the results clearly and completely presented?

-Are the figures (Tables, Images) of sufficient quality for clarity?

Reviewer #1: yes.

Reviewer #2: (No Response)

**Conclusions**

-Are the conclusions supported by the data presented?

-Are the limitations of analysis clearly described?

-Do the authors discuss how these data can be helpful to advance our understanding of the topic under study?

-Is public health relevance addressed?

Reviewer #1: yes.

Reviewer #2: (No Response)

**Editorial and Data Presentation Modifications?**

Reviewer #1: The author well responded the comments and I have no more comments.

Reviewer #2: (No Response)

**Summary and General Comments**

Reviewer #1: The author well responded the comments and I have no more comments.

Reviewer #2: (No Response)

PLOS authors have the option to publish the peer review history of their article (what does this mean? ). If published, this will include your full peer review and any attached files.

**Do you want your identity to be public for this peer review?** For information about this choice, including consent withdrawal, please see our Privacy Policy .

Reviewer #1: **Yes: ** Mya Myat Ngwe Tun

Reviewer #2: No

---

## [Editor Report · Acceptance letter]

Dear Professor ding,

We are delighted to inform you that your manuscript, "Development of monoclonal antibodies against E1 protein of Chikungunya virus," has been formally accepted for publication in PLOS Neglected Tropical Diseases.

Best regards,

Shaden Kamhawi

co-Editor-in-Chief

Paul Brindley

co-Editor-in-Chief
